# A Novel Cuticular Protein-like *Cpr21L* Is Essential for Nymph Survival and Male Fecundity in the Brown Planthopper

**DOI:** 10.3390/ijms24032163

**Published:** 2023-01-21

**Authors:** Tongtong Chen, Qiqi Jiao, Chenglong Ye, Jiangen Wu, Yuanyuan Zheng, Chuanxin Sun, Peiying Hao, Xiaoping Yu

**Affiliations:** 1Zhejiang Provincial Key Laboratory of Biometrology and Inspection & Quarantine, College of Life Sciences, China Jiliang University, Hangzhou 310018, China; 2Department of Plant Biology, Uppsala BioCenter, Swedish University of Agricultural Sciences, P.O. Box 7080, SE-750 07 Uppsala, Sweden

**Keywords:** *Nilaparvata lugens*, *Cpr21L*, RNA interference, survival, male fecundity

## Abstract

Cuticular proteins (CPs) are a large family and perform a variety of functions. However, the physiological roles of cuticle protein 21-like (*Cpr21L*) in the brown planthopper (*Nilaparvata lugens*, BPH), one of the most destructive insect pests of rice, are largely unclear. In this study, *Cpr21L* was revealed to be expressed in both BPH nymphs and adults, and the mRNA expression level was much higher in male adults than female adults. Spatially, the expression of *Cpr21L* in the testis was higher than in the ovary. The RNA interference (RNAi) of *Cpr21L* seriously decreased nymph survival, and no individual survived 8 days post-ds*Cpr21L* injection. The RNAi of *Cpr21L* in adults also decreased the fertility of males, especially in the ds*Cpr21L*♂ × ds*GFP*♀ group. The average number of eggs laid by one female in this group significantly decreased by 50.1%, and the eggs’ hatchability decreased from 76.5% to 23.8% compared with the control (ds*GFP*♂ × ds*GFP*♀). Furthermore, observations under a stereomicroscope showed that the RNAi of *Cpr21L* severely impaired the development of the testes. Therefore, *Cpr21L* is essential for the nymphal survival and male fecundity of BPH, thus providing a possible target for pest control.

## 1. Introduction

The brown planthopper (BPH), *Nilaparvata lugens* (Stål) (Hemiptera: Delphacidae), is a destructive insect rice pest [1]. The BPH depends on its piercing–sucking mouthparts to ingest sap from the phloem of rice. The feeding of a tremendous population of BPHs often causes huge losses of rice product. The management of BPHs mainly relies on chemical control, but the excessive and unscientific use of pesticides has resulted in frequent outbreaks of BPH and has caused potential environmental problems in recent decades [2]. Therefore, it is important to find an environmentally friendly way to control the BPH population by reducing their ability to survive and/or reproduce.

The insect cuticle is an extracellular structure that is mainly composed of cuticular proteins (CPs) and chitin. Cuticular proteins play a major role in determining the physical properties of the cuticle [3] and providing protection against microorganisms [4] and insecticides [5]. In recent years, a very large number of insect CPs and CP-like proteins have been identified and divided into different families according to their conserved protein sequence motifs [6]. In *Pteromalus puparum*, a total of 82 genes have been identified, and the encoding CPs have been grouped into 6 CP families [7]. In *Propsilocerus akamusi*, 160 CP genes have been identified, and 97 of them have been classified into 8 CP families [6]. In *Plutella xylostella*, 196 CP genes have been successfully annotated in its genome, and these CPs have been classified into 10 different families [8]. Therefore, the numbers and features of cuticular proteins are diversified among insects [7].

In BPH, a total of 140 CPs have been identified and classed into 8 CP families. Seventeen CPs have been revealed to be indispensable for normal nymph/adult development, among which the RNAi of 15 CP genes led to lethal phenotypes. In addition, 20 CPs play roles in egg production/embryo development, and the knockdown of 19 different CP genes in females significantly decreased egg hatchability [3]. However, the effect of CP genes on male fecundity in BPHs remains largely unknown. Recently, we found that the expression of a cuticle protein 21-like gene (*Cpr21L*) decreased upon the RNA interference of an autophagy-related gene, *ATG5*, in a transcriptome analysis of BPH, but the role of *Cpr21L* remains unclear. In this study, we cloned the ORF of *Cpr21L* and analyzed its spatiotemporal expression profiles. We also explored the function of *Cpr21L* in the BPH using RNAi techniques. Our results showed that the RNAi-mediated knockdown of *Cpr21L* seriously impaired the cuticle development and survival of BPH nymphs. In particular, the fecundity of adult males was severely impaired upon RNAi of *Cpr21L*. The present study provides insights into the physiological functions of *Cpr21L* in BPHs and provides new clues for pest control.

## 2. Results

### 2.1. Sequence and Phylogenetic Analysis

The ORF of the cuticle protein 21-like gene was cloned using the cDNA of BPH as the template for PCR. The sequencing result showed that the ORF of *Cpr21L* was 711 bp in length, encoding a peptide of 236 amino acids (XP_022193663.2). There was no signal peptide predicted in the Cpr21L protein, but one Chitin_bind_4 domain (aa 167-218) was predicted (Figure 1). The Chitin_bind_4 domain is usually found in the cuticle proteins and pro-resilins of insects and usually begins near a triad of aromatic residues (Y/F-x-Y/F/W-x-Y/F) and terminates shortly after a uniformly conserved G-F/Y12. Regarding Cpr21L, a predicted Chitin_bind_4 domain begins at residues YEFTY and terminates after residues GY (Figure 1). Thus, the Cpr21L was predicted to be a cuticle protein like.

As revealed by phylogenetic tree analysis with 10 other homologue protein sequences, the homologues of Cpr21L were distributed in different orders of insects (Hemiptera, Hymenoptera, and Coleoptera). Generally, most of the homologue proteins were clustered according to their order (Figure 2). Notably, the Cpr21L protein in BPH and a hypothetical protein RZF32538.1 (LSTR_LSTR011317, function unknown) in small brown planthopper (*Laodelphax striatellus*, SBPH) shared high similarity (identities, 77%; positives, 87%) and were distributed in the same branch. This is due to the fact that BPH and SBPH have a closer relationship than the other species analyzed. However, the Cpr21L and the hypothetical protein RZF32538.1 did not cluster in any other branch of Hemiptera, which included XP 0254166199.1 (*Sipha flava*), XP 026807875.1 (*Rhopalosiphum maidis*), XP 025199668.1 (*Melanaphis sacchari*), and KAF0761746.1 (*Aphis craccivora*), indicating their different ancestral origins. Therefore, the Cpr21L of BPH might represent a novel member of insect cuticular proteins, and its function needs to be further explored.

### 2.2. Spatiotemporal Expression Patterns of the Cpr21L Gene

RT-qPCR showed that *Cpr21L* was expressed in both nymphs and adults of the BPH. With the development of nymphs, the expression of *Cpr21L* increased from the 1st to the 2nd instar and reached a high level in the 5th instar nymphs. However, the expression of *Cpr21L* in female adults was generally low, except the first day post-emergence. In contrast to the female adults, the expression of *Cpr21L* was much higher in the male adults 1 to 5 days post-eclosion, and it showed a decreasing trend (Figure 3A). Spatially, the expression of *Cpr21L* in the head of the BPH was the highest, and it was relatively higher in the testis and epidermis than the other tissues or parts. Notably, the expression of *Cpr21L* in the ovary was the lowest, and it was only 3% of that in the head (Figure 3B).

### 2.3. Effects of Cpr21L Knockdown on the Survival and Development of BPH

To analyze the effect of *Cpr21L* knockdown on the BPH, nymphs of the 4th instar were injected with ds*Cpr21L*, using ds*GFP* as control. Three biological replicates were prepared, and each biological replicate contained 20 BPHs. The RT-qPCR results showed that the expression of the *Cpr21L* gene in the ds*Cpr21L*-treated group significantly decreased to 16.0% that of the ds*GFP* control at day 2 post-dsRNA treatment, indicating that RNAi had effectively knocked down the expression of the targeting gene *Cpr21L* (Figure 4A). As a result, the RNAi of *Cpr21L* significantly decreased the survival of the BPH from day 2 post-dsRNA treatment. Most importantly, almost all BPHs died at day 8. Meanwhile, the survival of the ds*GFP* control remained at higher levels (Figure 4B).

The RNAi of *Cpr21L* also affected the growth and development of the BPHs; for example, when some nymphs of the *GFP* control reached the 5th instar, some BPHs in the ds*Cpr21L*-treated group still stayed at the 4th instar (Figure 5A). At a later time, when some BPHs in the *GFP* group became adults, some of the ds*Cpr21L*-treated BPHs remained at the 4th or 5th instar (Figure 5B).

To investigate the function of the *Cpr21L* gene in adult BPHs, 20 male adults and 20 female adults within 12 h of emergence were injected with ds*GFP* or ds*Cpr21L*. One injected female and one injected male, as a pair, were allowed to mate freely on fresh rice seedlings. In total, 20 pairs were prepared. After 6 days of dsRNA-injection treatment, the BPH was observed under a stereomicroscope. The result showed that males injected with ds*Cpr21L* were shorter and thinner than those in the ds*GFP* control group. The abdomens of the ds*Cpr21L*-treated males in particular appeared more shriveled (Figure 5C). In contrast, the body size of ds*Cpr21L*-treated females did not change obviously when compared with the *GFP* control (Figure 5D). This indicates that *Cpr21L* played a more important role in male adults than it did in female adults.

### 2.4. Effects of Cpr21L Knockdown on the Structure of the Epidermis and Cuticle

To examine the effects of *Cpr21L* on the structure of the epidermis and cuticle of the BPH, RNAi was performed by injecting ds*Cpr21L* or ds*GFP* in 4th instar nymphs. Samples were taken 48 h later for TEM, and the results showed that there were many lamellar body-like (LBL) structures in the epidermal cells of BPHs injected with ds*GFP*; in contrast, there were fewer LBL structures in the BPHs injected with ds*Cpr21L* (Figure 6A–D,H). In addition, the volume of LBL structures in the ds*Cpr21L*-treated group were much smaller than those in the *GFP* control (Figure 6A–D,G), which strongly suggested that the inclusion of the Cpr21L protein in LBL structures was probably reduced. It should be noted that the cuticle of the ds*Cpr21L*-treated group was thinner than that of the *GFP* control group. In particular, the endocuticle of the BPH in the control group showed about 21 endocuticular lamellae (Figure 6E), while the endocuticle of the BPHs injected with ds*Cpr21L* only showed about 9 endocuticular lamellae (Figure 6F), which may be related to the decreased synthesis and secretion of Cpr21L protein.

### 2.5. RNAi of Cpr21L Impacted the Fecundity of Adults

To examine the effect of RNAi on the fecundity of BPH, newly emerged (within 12 h) male and female adults were separately injected with ds*Cpr21L* or ds*GFP*, and a pair of treated adults, one male and one female, were placed on a TN1 rice plant for mating and laying eggs. As a result, the average number of eggs laid by a female in a ds*Cpr21L*-treated male and ds*GFP*-treated female (ds*Cpr21L*♂ × ds*GFP*♀) pair significantly decreased by 50.1% compared with the control (ds*GFP*♀ × ds*GFP*♂) (Figure 7). Most importantly, the egg hatchability of the ds*Cpr21L*♂ group (ds*Cpr21L*♂ × ds*GFP*♀) was significantly reduced to 23.8%, but it remained at a high level of 76.5% in the control (ds*GFP*♀ × ds*GFP*♂). However, egg hatchability did not show any significant differences between the ds*Cpr21L*♀ × ds*GFP*♂ pair and the *GFP* control (Figure 8A).

### 2.6. RNAi of Cpr21L Resulted in Malformed Internal Reproductive Organ of Male Adult

To analyze how the RNAi of *Cpr21L* impacted the fecundity of male adults, the morphology of the internal reproductive organ of male adults was examined. The results showed that the testes of the males injected with ds*GFP* developed well, and each testis displayed 3 pepper-shaped branches (Figure 9a,c). In contrast, the males injected with ds*Cpr21L* were abnormal in the shape of the testes, and the testes did not display any pepper-shaped branches. Instead, the branches of the testes treated with ds*Cpr21L* looked like small bubbles or sacs (Figure 9b,d). In addition, the vas deferentia of the males in the ds*Cpr21L* group showed some nodules. The results showed that the RNAi of the *Cpr21L* gene had a strong impact on the development of the internal reproductive organs of male BPH, which particularly affected the development of the testes (Figure 9d). However, no clear morphologic difference in the internal reproductive organs was observed between females treated with ds*Cpr21L* and those treated with ds*GFP*.

## 3. Discussion

Cuticular proteins serve several roles in the development and reproduction of insects. Although cuticular proteins have been studied in several insects, the function and importance of the cuticular protein-like Cpr21L in BPH remains largely unknown. In this work, we revealed that *Cpr21L* is highly expressed in both nymphs and male adults but lowly expressed in female adults. Correspondingly, the RNAi of *Cpr21L* resulted in the high mortality of nymphs and severely impaired the fecundity of male adults. We found that the RNAi of *Cpr21L* clearly affected the development of the testis, indicating that *Cpr21L* affected the fecundity of male adults, probably by impairing the development and function of the testes. Previous studies on the role of cuticular proteins in reproduction have mainly focused on females [3], but there are few reports on their roles in the fecundity of males. Therefore, our study provided new insights into the function of cuticular proteins such as Cpr21L on the fecundity of male BPHs.

The insect cuticle is composed of many kinds of cuticular proteins together with chitin. The insect cuticle consists of three horizontal layers (epicuticle, exocuticle, and endocuticle), which are secreted by epidermal cells [9]. In this study, TEM observations indicated that the cuticle (endocuticle) of ds*Cpr21L*-treated nymphs was thinner than that of the *GFP* control, and there were fewer and smaller LBL structures in the epidermal cells of the former. Although we have not identified the exact content of the LBL structures at present, the available evidence encourages us to speculate that it might be some secretory vesicles containing Cpr21L protein. Given that the Cpr21L protein was in the LBL structures, enough protein can be successfully secreted by the epidermal cell mediated by the secretory LBL structures under normal conditions. On the contrary, the RNAi of *Cpr21L* resulted in less synthesis and secretion of Cpr21L protein and a thinner cuticle. Finally, the BPH nymphs treated with ds*Cpr21L* die of an impaired cuticle caused by less Cpr21L protein synthesis and secretion. In eukaryotes, the majority of secretory proteins are secreted via conventional secretion. These secretory proteins generally contain a signal peptide which can be recognized and bound by the signal recognition particle (SRP), and subsequently are transferred to the endoplasmic reticulum (ER) through the translocon [10,11,12]. The secretory proteins are then exported through ER–Golgi trafficking vesicles [13,14]. However, Cpr21L lacks a classical signal peptide, and how it is secreted needs to be explained. Recently, it was found that some proteins without a classical signal peptide can also be secreted via a non-conventional secretion pathway, and this kind of protein contains sequences named motif-1 and motif-2. Notably, motif-1 has been proven to be sufficient to direct a protein lacking a classical signal peptide to the route of secretion [15]. Interestingly, we also found two sequences sharing some similarity with motif-1 and motif-2 in Cpr21L (Appendix A), strongly suggesting that Cpr21L is probably secreted through a non-conventional secretion pathway rather than a conventional secretion pathway. Therefore, our work provides clues for further exploring the mechanism of how Cpr21L is secreted by the epidermal cell [15].

As is described above, *Cpr21L* is highly expressed in the cuticle, and so is reasonable to speculate that Cpr21L binds with chitin through its Chitin_bind_4 domain. However, no chitin has been reported to exist in the testis, so why is *Cpr21L* also highly expressed in the testis? There are two possibilities: First, the interaction of Cpr21L and chitin is related to the development of the testis. For example, the oral administration of Etoxazole (ETX, a pesticide that specifically inhibits chitin synthesis) in C57BL/6 male mice reduced testis weight and altered transcriptional expression related to testis function [16]. In addition, Pan et al. (2018) also identified a testis protein family TPAP (testis proteins analogous to peritrophins) in BPH that contains a chitin-binding domain [3], suggesting the possible existence of chitin in the testis [16]. Second, Cpr21L might be a multifunction protein that does not bind chitin in the testis or internal reproductive organ of the male. In fact, Cpr21L shared moderate similarity with many pro-resilins in insects, indicating that Cpr21L may play roles similar to those of resilins. Generally, resilin is responsible for the elasticity of insect integumental structures, especially at articulations, such as wing hinges and tendons [17,18]. Recently, resilin was found in the sperm pump of *Monotoma*, and it is thought to function in transferring seminal fluids and sperm from the testes into the aedeagus by shrinking and expanding [19]. Therefore, Cpr21L in the internal reproductive organ of a male might play roles in the development of the testis and function in transferring sperm from the testes into the aedeagus by shrinking and expanding. Further studies are needed to elucidate whether there is some chitin in the testes and how Cpr21L affects the development of the testes. In addition, the expression of *NlCpr21L* was high in the head of the BPH. As described above, NlCpr21L shared moderate similarity with many pro-resilins in insects, indicating that NlCpr21L may play roles similar to those of resilins. Rebora et al. (2021) analyzed the ultrastructure and development of the white patches on the head and thorax of *Bactrocera oleae* and found that the white patches also show UV-induced blue autofluorescence due to the air sac resilin content [20]. Therefore, NlCpr21L in the head of BPH may play a similar role to resilin in the white patches on the head of *B. oleae*. Therefore, further research on the function of NlCpr21L in the head is required.

In summary, our findings demonstrate the important role of Cpr21L in the survival of nymphs and the fecundity of adult males and provide a potential target gene, *Cpr21L*, for BPH control. However, a deeper understanding of how Cpr21L affects the function of the testis is needed.

## 4. Materials and Methods

### 4.1. Insects

The BPH insects used in this study were reared on TN1 rice in China Jiliang University, Hangzhou, China. The insect and rice were maintained in a growth chamber at 26 ± 2 °C, relative humidity 70 ± 5%, and a 16 h/8 h (light/dark) photoperiod.

### 4.2. RNA Isolation

Total RNA was extracted from the first to fifth instar nymphs, females and males, 1, 3, 5, 7, and 9 days post-emergence. For each stage, three replicates were prepared using 10 adults or 30–50 nymphs per replicate. The insect body was cleaned with 75% alcohol quickly and washed with 1× BPS. The head and thorax of the BPH were cut and collected under a stereomicroscope with a scalpel. The abdomen was torn out with dissecting forceps, and the midgut or fat body was collected separately. Similarly, the ovary was obtained from the abdomen of the female, and the testis was obtained from the abdomen of the male. The epidermis was collected after the midgut, fat body, and ovary or testis were removed. RNA was extracted using the TRIzol reagent (Invitrogen, Carlsbad, CA, USA) following the manufacturer’s instructions.

### 4.3. cDNA Synthesis

The concentration of total RNA was measured using a NanoDrop 2000 spectrophotometer (Thermo Fisher Scientific, Waltham, MA, USA). A volume of 1.0 μg of total RNA was calculated, and then the first-strand cDNA was synthesized using the PrimeScript RTReagent Kit with gDNAEraser (Takara, Tokyo, Japan). We followed the manufacturer’s protocols for subsequent PCR and RT-qPCR.

### 4.4. Sequence and Phylogenetic Analysis of the Cpr21L in BPH

The nucleotide sequences of the *Cpr21L* gene (GenBank accession number: LOC111051450) were acquired from the transcriptome data of BPH, in which an autophagy gene, *ATG5*, was knocked down using RNA interference in our lab (unpublished data). The open reading frame (ORF) of *Cpr21L* was verified by sequencing the RT-PCR product with the primers in Table 1. The primers were designed using Primer Premier 5.0 to clone the ORF of *Cpr21L*. The PCR system contained 50 µL, including 25 µL of Premix TaqTM (Takara, Tokyo, Japan), 2 µL of each primer (10 μM each), 2 µL of cDNA template, and 19 µL of ddH2O. The PCR procedure was as follows: 95 °C for 3 min; 34 cycles of 95 °C for 30 s, 60 °C for 30 s, and 72 °C for 30 s; and finally 72 °C for 3 min. After the PCR was completed, PCR product in strips of the expected size was collected using a MiniBESTAgarose Gel DNAExtraction Kit (Takara, Tokyo, Japan), ligated to the pMD18-T vector (Takara, Tokyo, Japan), and then transformed into cells of *Escherichia coli* DH5α. Colonies that were confirmed to have transformed the target bands were selected and sequenced at Zhejiang Youkang Biotechnology Co., Ltd. (Hangzhou, China). The signal peptide sequence was predicted by the SignalP 5.0 service. (https://services.healthtech.dtu.dk/service.php?SignalP-5.0/) (accessed on 5 December 2022). The domains of Cpr21L were predicted using SMART (http://smart.embl-heidelberg.de/) (accessed on 15 January 2023). Orthologs of Cpr21L in different insects were searched using the BLASTP algorithm on the NCBI website. A phylogenetic tree was constructed using MEGA 7.0 (https://www.megasoftware.net/) (accessed on 15 January 2023) software via the neighbor-joining method with 1000 bootstrap replications.

### 4.5. Spatiotemporal Expression Patterns of Cpr21L

In this part of the experiment, the previously obtained cDNA was used to carry out experiments using the StepOnePlus Real-Time PCRSystem (Applied Biosystem, Foster City, CA, USA) to explore the time-specific and tissue-specific expression patterns of *Cpr21L* in the BPH. Primers (Table 1) for RT-qPCR were designed using Primer Premier 5.0. The 20-µL RT-qPCR reaction system contained 10 µL of TBGreen Rremix Ex Taq II (Takara, Tokyo, Japan), 0.4 µL of each primer (10 μM), 2 µL of cDNA template (four-fold dilution), 0.4 µL of ROXReference Dye, and 6.8 µL of H2O. The PCR conditions were 95°C for 30 s; 40 cycles of 95 °C for 5 s and 60 °C for 30 s; and 95 °C for 15 s, 60 °C for 60 s, and 95 °C for 15 s. The *RPS11* of the BPH was used as an internal control (Table 1) [21]. Three biological replicates were used in this experiment, and each one included three technical replicates. The relative expression levels of genes were evaluated using the 2−ΔΔCt method [22].

### 4.6. Double-Stranded RNA Synthesis and Microinjection

Primers were designed for synthesizing dsRNA using Primer Premier 5.0 based on the sequence of *Cpr21L*. A green fluorescent protein GFP (GenBank: MF169984.1) was used as the control. The primers (Table 1) for ds*GFP* and ds*Cpr21L* were used to synthesize the double-stranded RNA (dsRNA), according to the protocols of the manufacturer of the MEGAscript^®^ T7 High Yield Transcription Kit (Ambion, Austin, TX, USA). Next, the concentration of the dsRNA was measured on a NanoDrop 2000 spectrophotometer.

Before injection, needles for the microinjections were made by pulling out 10 µL calibrated pipets (Drummond Scientific Company, Broomall, PA, USA) with a PC-10 microelectrode puller (Narishige, Tokyo, Japan), and dsRNA was transferred to the needles with a pipette gun. The BPHs were injected using a capillary tube at the base of the 2nd or 3rd leg under the control of a FemtoJet 4i microinjector (Eppendorf, Hamburg, Germany). The selected BPHs were frozen in advance on ice to facilitate injection. Finally, after injection, the revived BPHs were gently transferred to fresh TN1 rice plants for feeding.

### 4.7. Effects of RNAi against Cpr21L on the Survival and Development of BPH

To observe the effects of RNAi against *Cpr21L* on the survival and development of the BPH, each fourth instar nymph was injected with 250 ng ds*Cpr21L* as the treated group, and the nymphs in the control group were injected with the same amount of ds*GFP*. Three biological replicates were prepared, and each biological replicate contained 20 BPHs. The number of surviving BPHs was recorded every 24 h. The development of the BPHs was observed and photographed with a Nikon C-DSS230 stereomicroscope (Nikon, Tokyo, Japan).

### 4.8. Transmission Electron Microscopy (TEM) of dsRNA-Treated BPHs

After having been treated with ds*GFP* or ds*Cpr21L* for 2 days, five nymphs of the 4th instar were sampled for each group and placed on ice, and their heads and thoraces were quickly removed with tweezers under a Nikon C-DSS230 stereomicroscope. The remaining part of the insect body without the head and thorax was quickly soaked in 2.5% glutaraldehyde solution and fixed overnight at 4 °C. After the samples were fixed well, subsequent processing was performed using a previously reported method [8,23]. Semi-thin sections (2 μm) were cut using glass knives on an LKB Bromma 11,800 pyramitome (LKB, Bromma, Sweden) and stained with methylene blue. The ultra-thin sections were prepared with a diamond knife using PowerTome-PC (RMC, Boeckeler Instruments, Tucson, AZ, USA). The sections were stained with 3% uranyl acetate and alkaline lead citrate and were observed using a Hitachi H-7650 TEM (Hitachi, Tokyo, Japan).

### 4.9. Effect of RNAi on Insect Fecundity

To examine the effect of RNAi on the fertility of BPHs, newly emerged (1–12 h) female adults and male adults were injected with ds*Cpr21L*. ds*GFP* was used as the control. The three groups were set as: ds*GFP*♀ × ds*GFP*♂, ds*Cpr21L*♂ × ds*GFP*♀, and ds*Cpr21L*♀ × ds*GFP*♂. Each group contained 20 pairs of adult insects. In each pair, one female and one male were allowed to mate freely on fresh rice seedlings (about 6 cm tall) in a glass tube (length =116 mm, diameter = 28 mm). After 3 days’ feeding, the BPHs were transferred to fresh rice seedlings in another glass tube. Rice seedlings were kept in each glass tube for 10 days, and the number of hatched offspring was calculated. Finally, rice seedlings were dissected under a stereomicroscope to count the number of eggs that failed to hatch.

### 4.10. Statistical Analysis

SPSS 22.0 (http://www.spss.com) (accessed on 15 January 2023) was used for the statistical analysis of the experimental data. Two groups were compared using the Student’s *t*-test, and differences among three or more groups were analyzed with one-way ANOVA, followed by Tukey’s multiple comparison test.

## Figures and Tables

**Figure 1 ijms-24-02163-f001:**
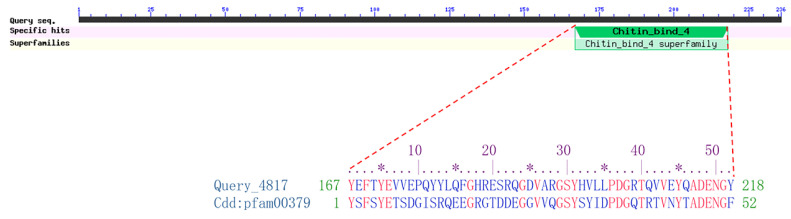
Conserved domain Chitin_bind_4 of Cpr21L. Alignment of Cpr21L (Query _4817) with conserved domain Cdd:pfam00379. Asterisk (*) means the 5th amino acid residue among 10 amino acid residues.

**Figure 2 ijms-24-02163-f002:**
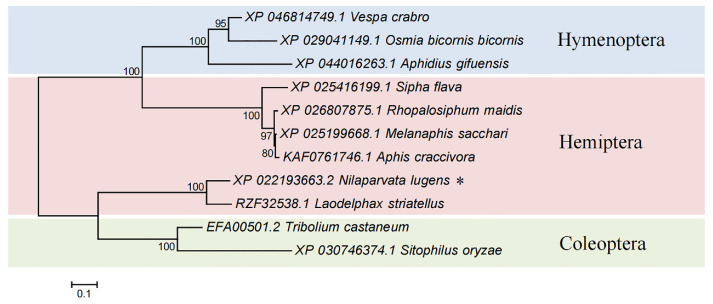
Phylogenic analysis of Cpr21L. The tree was constructed using MEGA 7.0 software (neighbor-joining method with *n* = 1000 bootstrap replications). Bootstrap values above 70% are shown on each node. Cpr21L (XP _022193663.2) is marked with an asterisk (*).

**Figure 3 ijms-24-02163-f003:**
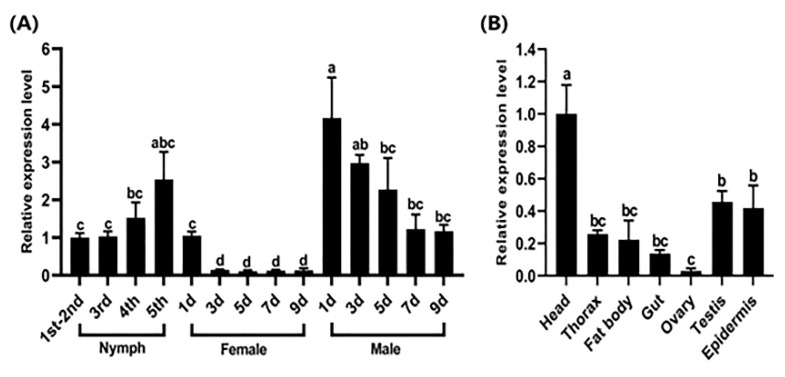
Spatiotemporal expression patterns of *Cpr21L*. (**A**) Expression patterns of *Cpr21L* in BPHs at different developmental stages. Samples were collected from nymphs (1st–5th instar), females, and males (1, 3, 5, 7, and 9 days post-eclosion). (**B**) Expression patterns of *Cpr21L* in different tissues or body parts. *RPS11* was used as the reference gene to normalize gene expression. Bars are the mean ± SEM (standard error of mean) from three independent experiments. The data were analyzed by one-way ANOVA and Tukey’s multiple comparison tests. Different letters "a", "b", and "c" on the bars indicate significant differences (*p* < 0.05), while the same letter were not significantly different (*p* < 0.05).

**Figure 4 ijms-24-02163-f004:**
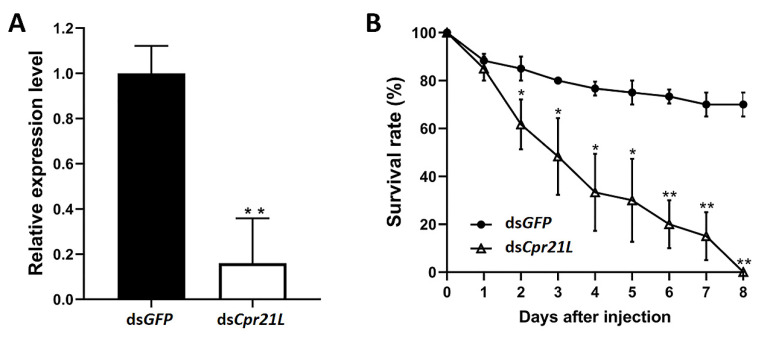
The effects of RNAi against *Cpr21L* in the BPH. (**A**) The mRNA level of *Cpr21L* in the 5th instar nymphs after being injected with ds*Cpr21L*. ds*GFP* was used as the control, and *RPS11* was used as the reference gene to normalize gene expression. (**B**) The survival of nymphs treated with ds*Cpr21L*. ds*GFP* served as a control; *n* = 40 insects. Values are expressed as the mean ± SEM of at least three independent biological replicates. The Student’s *t*-test was used to compare two groups of samples at the same time point. (* *p* < 0.05; ** *p* < 0.01).

**Figure 5 ijms-24-02163-f005:**
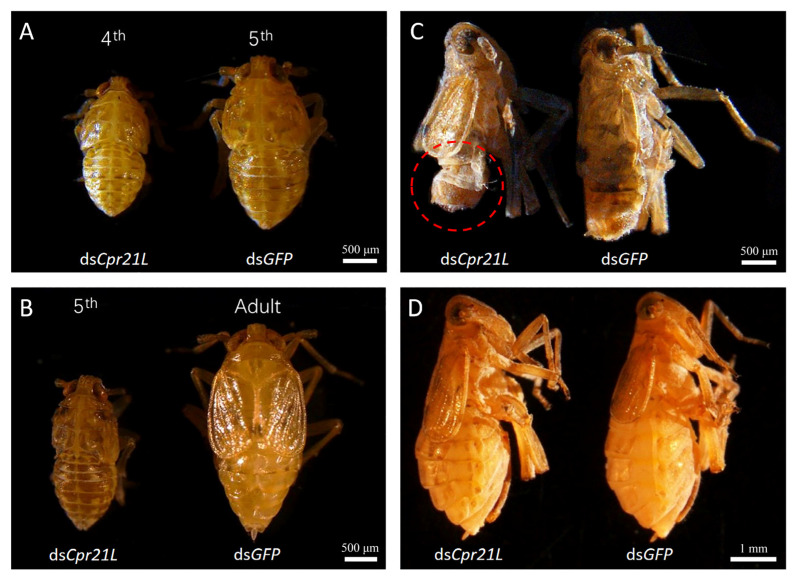
Effects of *Cpr21L* knockdown on the development of BPHs. (**A**,**B**) Effect of RNAi against *Cpr21L* on the growth and development of BPH nymphs. (**C**) Effect of RNAi of *Cpr21L* on the male adult. (**D**) Effect of RNAi of *Cpr21L* on the female adult. Scale bars: (**A**), (**B**), and (**C**) = 500 μm; (**D**) = 1 mm.

**Figure 6 ijms-24-02163-f006:**
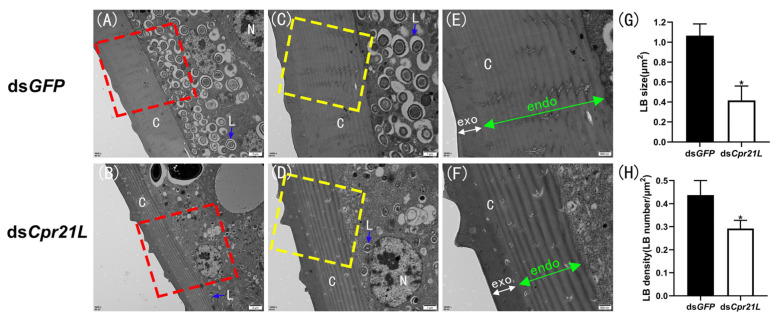
Transmission electron microscopy (TEM) observations of the abdominal cuticle of dsRNA-injected BPHs. Nymphs were injected with dsRNAs at the 4th instar stage (1–12 h post-molting) and then sampled 2 days later. (**C**) and (**D**) are magnifications of the areas marked with red-dashed boxes in (**A**) and (**B**), respectively. Likewise, (**E**) and (**F**) are magnifications of the areas marked by the yellow-dashed boxes in (**C**) and (**D**), respectively. Blue arrows, lamellar body-like (LBL) structures. Scale bar = 2 μm (**A**,**B**), 1 μm (**C**,**D**), and 500 nm (**E**,**F**). (**G**) Size of LBL structures. (**H**) Density of LBL structures. Asterisks (*) on the bars in (**G**) and (**H**) indicate significant differences between the control (dsGFP) and the corresponding treatment group (* *p* < 0.05).

**Figure 7 ijms-24-02163-f007:**
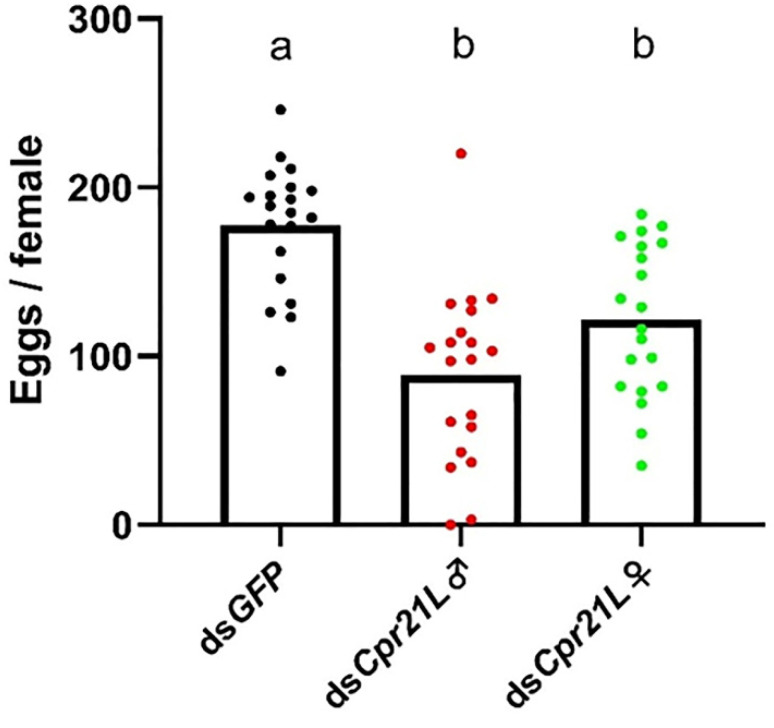
The effects of RNAi on the average number of eggs laid by one female BPH. Newly emerged females and males (1–12 h) were separately injected with ds*GFP* or ds*Cpr21L* and then mated 24 h later. ds*GFP* (control): ds*GFP*♀ × ds*GFP*♂; ds*Cpr21L*♂: ds*Cpr21L*♂ × ds*GFP*♀; ds*Cpr21L*♀: ds*Cpr21L*♀ × ds*GFP*♂. Bars are the mean ± SEM from 20 females. The data were analyzed using one-way ANOVA, followed by Tukey’s multiple comparison test. Different letters "a" and "b" on the bars indicate significant differences (*p* < 0.05), while the same letter were not significantly different (*p* < 0.05)..

**Figure 8 ijms-24-02163-f008:**
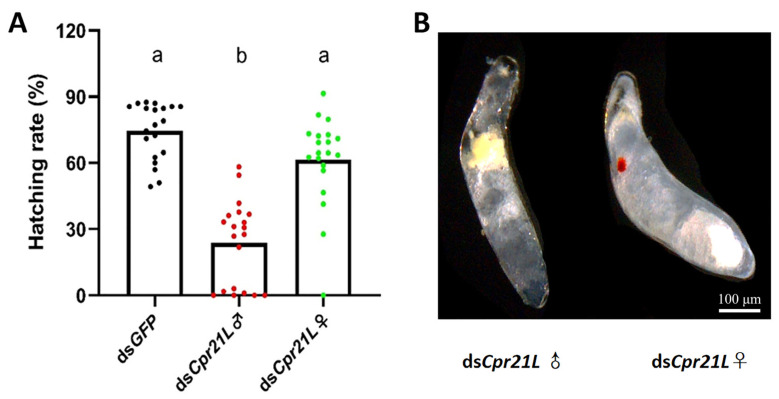
The effects of RNAi on the egg development and hatchability of BPH. (**A**) Egg hatching rate. (**B**) Eggs laid by the BPH and isolated from the rice leafsheath. Newly emerged males and females (1–12 h) were separately injected with ds*Cpr21L* or ds*GFP* (control), and 6 days later the eggs were isolated. ds*GFP* (control): ds*GFP*♀ × ds*GFP*♂; ds*Cpr21L*♂: ds*Cpr21L*♂ × ds*GFP*♀; ds*Cpr21L*♀: ds*Cpr21L*♀ × ds*GFP*♂. Bars are the mean ± SEM from 20 females. The morphology of 6-day-old eggs was observed. Scale bar = 100 μm. Different letters "a" and "b" on the bars indicate significant differences (*p* < 0.05), while the same letter were not significantly different (*p* < 0.05). In normal situations, involving a normal egg, normal sperm, and successful fertilization, the fertilized egg or embryo will develop well, and red-eye spots are expected to occur at the 5th day post oviposition. To investigate why the eggs’ hatchability in the ds*Cpr21L*♂ × ds*GFP*♀ group was low, the eggs were isolated from the rice leafsheath and observed under a stereomicroscope. The result showed that most of the eggs (embryo) from the ds*Cpr21L*♀ × ds*GFP*♂ group showed obvious red-eye spots, suggesting that the RNAi of the ds*Cpr21L* did not affect the quality of the eggs in the ds*Cpr21L* treated females; therefore, if these eggs were well fertilized, the fertilized eggs were expected to develop normally, and nymphs could hatch from these eggs (Figure 8B, right). On the contrary, most of the eggs from the ds*Cpr21L*♂ × ds*GFP*♀ group did not show red-eye spots, so it is possible that these eggs were unfertilized (Figure 8B, left). This strongly suggests that the RNAi of ds*Cpr21L* affects the quality and/or quantity of the sperm in the ds*Cpr21L*-treated males.

**Figure 9 ijms-24-02163-f009:**
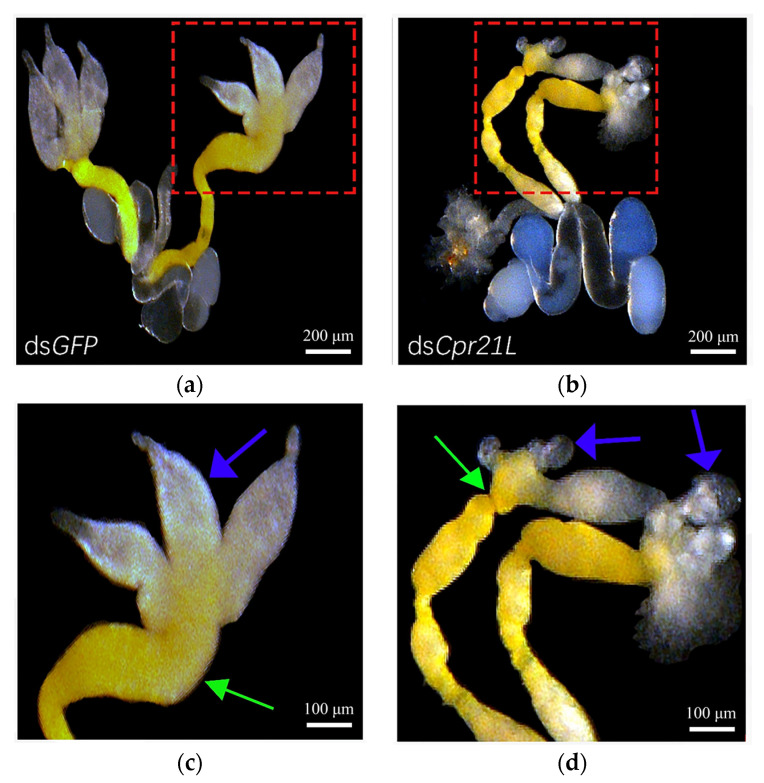
The effects of RNAi on the morphology of the internal reproductive organs of male adults. Newly emerged males (1–12 h) were injected with ds *Cpr21L*, or ds*GFP* (control), and 6 days later their internal reproductive organs were dissected. (**c**) and (**d**) are magnifications of the area marked with the red-dashed boxes in (**a**) and (**b**), respectively. The branches of the testes are indicated by blue arrows, and the vas deferentia are indicated by green arrows.

**Table 1 ijms-24-02163-t001:** List of primers used in this study.

Primer Name	Sequence (5’→3’)
for ORF cloning	
*Cpr21L*-F	CAAGTCCACGGTGTGTTGTG
*Cpr21L*-R	TTAATATCGATTATCGTAGGC
for qPCR	
*Cpr21L*-qF	GAGATGTGGCGCGTGGAT
*Cpr21L*-qR	GCTGGCTGAGCTTTTTGC
Nl*RPS11*-qF	CCGATCGTGTGGCGTTGAAGGG
Nl*RPS11*-qR	ATGGCCGACATTCTTCCAGGTCC
for dsRNA synthesis	
ds*Cpr21L*-F	GGATCCTAATACGACTCACTATAGGGTGTGTTGTGAACTTGTGAAA
ds*Cpr21L*-R	GGATCCTAATACGACTCACTATAGGGCTCCCTGTAAGAATTGTCTG
ds*GFP*-F	GGATCCTAATACGACTCACTATAGGGAAGGGCGAGGAGCTGTTCACCG
ds*GFP*-R	GGATCCTAATACGACTCACTATAGGGCAGCAGGACCATGTGATCGCGC

## Data Availability

Not applicable.

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
