# Peer review of "A Novel Cuticular Protein-like Cpr21L Is Essential for Nymph Survival and Male Fecundity in the Brown Planthopper"

_ijms, 2023, doi:10.3390/ijms24032163_

Round 1
Reviewer 1 Report
The brown planthopper (BPH) is one of the most devastating insect pests of rice. It is important to control the infestation of brown planthopper. This manuscript reports a promising target Cpr21L for pest control, which is essential for nymphal survival and male fecundity of BPH. The design and interpretation are good. I would recommend a few minor revisions.
1. Line 52-53, Authors chose Cpr21L gene as a candidate, for its expression decreased upon the RNAi of an autophagy-related gene ATG5 in the transcriptome analysis of BPH. Because the transcriptome data hasn’t been published, it would be better, that some information on the transcriptome was uploaded as a supplementary material (for review only).
2. Line 121-122, The BPHs were injected under the control of a FemtoJet 4i microinjec-121 tor, but it was not clear against which part (thorax, or abdom?), the microinjection was performed? If the microinjection was performed against the abdom, it is possible that the microinjection will damage the ovary or testis.
3. Figure 3, It showed that the expression of NlCpr21L was high in the head, but there was no description on the effect of RNAi on the head. I suggest authors give some explain about this in the Discussion section.
4. In figure 3. The gene expression of NlCpr21L in the females was in fact not high, why the dsRNA injection was performed on females?
5. Scale bar should be added to Figure 8 B.
6. Line 17, Change “RNAi-mediated knockdown” to “RNAi”.
7. Line 334, study on the role of cuticular proteins in reproduction mainly focused on females3, change “females3” into “females[3]”. Similarly, in Line 369-370, “reduced testis weight and altered transcriptional expression related to testis function.21” should be changed into “reduced testis weight and altered transcriptional expression related to testis function.[21]”
8. Some References, such as Reference 1, appears only one page. Make sure to correctly cite the references, providing full page information.
Reviewer 2 Report
The MS and the work is excellent which shows role of Cpr21L in survival BPH nymph and the fecundity of male adult, which could be a potential target gene Cpr21L 385 for BPH control

Reviewer 3 Report
Brief summary
In this manuscript the authors were exploring the cuticular protein 21-like (Cpr21L) and its physiological roles in the nymphs and adults of brown planthopper Nilaparvata lugens. The selection of the insect species was appropriate as this is one of the most destructive pests in rice.
The manuscript was written well and easy to understand even for those who are not as familiar with the subject. The authors provided good background and introduction to the study. The methodology was explained in such a way that outlined what were done in the studies although this section needs some improvement – see specific comments below. The results were presented in graphs format and photographs which tremendously helpful to visualize each data presented. The discussion section strengthened the results section by showing previous studies done that supported the current studies. The authors concluded the study based on their findings that Cpr21L was important in nymph and male fecundity of the brown planthopper (BPH) and suggested this gene for BPH control.
Specific comments:
There are various formatting issues (primarily in the references section) within the document which can be easily corrected. See details below.
Introduction (page 2)
Lines 53
… RNA interfere (RNAi) … à should be RNA interference
Materials and Methods (pages 2-4)
Lines 64-67
The insect rearing and plant (rice) growth conditions were mentioned. However, there was no information on where and how that was done. Was it in greenhouse? Or was it in growth chamber? Did you follow a methodology that was previously done, and if so, could you provide a reference (or references)? In other words, more details need to be provided in this section.
Lines 68-72
For the RNA extraction, how did each body parts of insects were separated? More information is needed o provide a reference if following published methodology.
Line 83
Opening reading frame (ORF) … à should be open reading frame
Lines 97-98
Provide a reference for MEGA 7.0 software
Lines 136-137
How many of the 4th instar nymphs were sampled? This needs to be mentioned as part of the methodology.
Line 141
Format for reference (11) needs to be corrected following journal guidelines.
Lines 141-142
The specimen sections were observed … à What are the sections that were observed? Provide details.
Line 154
Provide reference for SPSS 22.0 software.
Results (pages 4-10)
Line 206
… mean ± SEM … à what is SEM? Spell out for the first time then abbreviate throughout.
Lines 210-217
How many nymphs were injected with dsCpr21L? Provide information.
Lines 223-231
How many samples of female and male adults were injected? Provide information.
Lines 278-283
Provide background information regarding red-eye spots. Is a red-eye spot an indication of something? Why red-eye spots suggesting no effect on egg quality if dsCpr21L treated females?
Line 282
…, therefor, … à should be therefore
… theses … à should be these
Discussion (pages 10-11)
I may have missed it as reviewing the paper but I could not find reference (20) listed anywhere in the document except in References list. It seems to be missing.
Line 359 and line 370
References (19) and (21) need to be formatted per IJMS guidelines.
Line 370
… Pan et al … à shouldn’t the year of publication be listed as in: Pan et al (2021)?
Line 375
… may paly … à should be: may play …
Line 378
Monotoma should be italicized (as in Monotoma)
References (pages 12-13)
Lines 406-457
Check with the IJMS guidelines and make correction throughout this section as I found there were inconsistencies throughout on formatting. For example, some journals are abbreviated and some are not; some years are in bold and some are not.
Another problem: A few references were not even mentioned the title of the journals. For example, reference 4.
Be sure species are italicized – there are many in this section that are not. For example, reference 7: Bombyx mori à should be Bombyx mori.
